# Stroke volume and cardiac output during 6 minute-walk tests are strong predictors of maximal oxygen uptake in people after stroke

**Fang Liu**[1]☉, **Alice Y. M. Jones**[2]☉*, **Raymond C. C. Tsang**[3]☉, **Fubing Zha**[1]‡, **Mingchao Zhou**[1]‡, **Kaiwen Xue**[4]‡, **Zeyu Zhang**[4]‡, **Yulong Wang**[1]☉*

**1** Department of Rehabilitation, Shenzhen Second People's Hospital, The First Affiliated Hospital of Shenzhen University Health Science Centre, Shenzhen, China, **2** School of Health and Rehabilitation Sciences, The University of Queensland, Queensland, Australia, **3** Department of Physiotherapy, MacLehose Medical Rehabilitation Centre, Hong Kong, China, **4** School of Rehabilitation Sciences, The Shandong University of Traditional Chinese Medicine, Shandong, China

☉ These authors contributed equally to this work.
‡ These authors also contributed equally to this work.
* a.jones15@uq.edu.au (AYMJ); ylwang668@163.com (YW)

**Data Availability Statement:** All relevant data are within the manuscript.

**Funding:** We confirm that our project was funded by the Sanming Project of Medicine in Shenzhen.

## Abstract

### Background and objectives

The 6-minute walk test (6MWT) is a field test commonly used to predict peak oxygen consumption ($VO_{2peak}$) in people after stroke. Inclusion of cardiodynamic variables measured by impedance cardiography (ICG) during a 6MWT has been shown to improve prediction of $VO_{2peak}$ in healthy adults but these data have not been considered in people after stroke. This study investigates whether the prediction of $VO_{2peak}$ can be improved by the inclusion of cardiovascular indices derived by impedance cardiography (ICG) during the 6MWT in people after stroke.

### Methods

This was a cross-sectional study. Patients diagnosed with stroke underwent in random order, a maximal cardiopulmonary exercise test (CPET) and 6MWT in separate dates. Heart rate (HR), stroke volume (SV) and cardiac output (CO) were measured by ICG during all tests. Oxygen consumption was recorded by a metabolic cart during the CPET. Recorded data were subjected to multiple regression analyses to generate $VO_{2peak}$ prediction equations.

### Results

Fifty-nine patients, mean age 50.0±11.7 years were included in the analysis. The mean distance covered in the 6MWT (6MWD) was 294±13 m, $VO_{2peak}$ was 19.2±3.2 ml/min/kg. Mean peak HR, SV and CO recorded during 6MWT were 109±6 bpm, 86.3±8.8 ml, 9.4±1.2 L/min and during CPET were 135±14 bpm, 86.6±9 ml, 11.7±2 L/min respectively. The prediction equation with inclusion of cardiodynamic variables: 16.855 + (-0.060 x age) + (0.196 x BMI) + (0.01 x 6MWD) + (-0.416 x $SV_{6MWT}$) + (3.587 x $CO_{6MWT}$) has a higher squared

Grant number SZSM202111010. The funders had no role in study design, data collection and analysis, decision to publish, or preparation of the manuscript.

**Competing interests:** The authors have declared that no competing interests exist.

multiple correlation ($R^2$) and a lower standard error of estimate (SEE) and SEE% compared to the equation using 6MWD as the only predictor.

## Conclusion

Inclusion of SV and CO measured during the 6MWT in stroke patients further improved the VO$_{2peak}$ prediction power compared to using 6MWD as a lone predictor.

## Introduction

Neurological deficits, a sedentary lifestyle and impaired diastolic function are all factors which impair exercise capacity of people after stroke [1,2]. Peak oxygen consumption (VO$_{2peak}$) measured after stroke is usually significantly lower than in sex-matched sedentary healthy individuals [3–5]. Suboptimal exercise capacity not only leads to low quality of life but also results in increased risks of recurrent stroke and accident [1,6].

Therapeutic exercise training is known for its potential in increasing aerobic capacity, an essential element for functional recovery in people after stroke [7,8]. Appropriate and effective prescription of a rehabilitation exercise program is guided by accurate assessment and prediction of the participant's aerobic capacity. Aerobic capacity, either predicted or from direct measurement, is an essential outcome for progress monitoring and program evaluation. Progressive cardiopulmonary exercise testing (CPET) is the gold standard for assessment of aerobic capacity [9]. However, in people after stroke, neuro-motor impairment, poor balance, and spasticity often limit them from reaching their maximum aerobic capacity [10].

The 6-minute walk test (6MWT) is a common field test used to monitor and evaluate submaximal aerobic capacity in people with chronic cardiopulmonary disease [11,12], and after stroke [13,14]. The outcome used in a 6MWT is the distance covered during this submaximal test (6MWD) and reflects the VO$_{2peak}$ (aerobic capacity) of the participant. However, the 'distance' covered during the test, particularly in the stroke population, can be influenced by non-hemodynamic factors. Heart rate (HR), stroke volume (SV) and cardiac output (CO) are the primary contributing factors to exercise capacity [15,16]. These cardiodynamic parameters can now be measured non-invasively and conveniently during a 6MWT. Information from these 'direct' cardio-dynamic factors could more reliably apprise the aerobic capacity of the participant, compared to using only the 6MWD as the outcome. Further, the predictive power of 6MWD for VO$_{2peak}$ in the stroke population has been questioned [10]. Inclusion of these parameters during a 6MWT for prediction VO$_{2peak}$ has recently been shown to be more accurate in predicting VO$_{2peak}$ than using 6MWD alone in young adults [17]. Whether such a relationship can be extended to people after stroke has not been determined.

This study aimed to explore, in a cohort of patients after stroke, (1) the relationship between cardiodynamic parameters (HR/SV/CO) and VO$_{2peak}$ recorded during a CPET and a 6MWT, and (2) whether inclusion of cardiodynamic parameters recorded during a 6MWT in the linear regression equation for prediction of VO$_{2peak}$ can further improve the accuracy and stability of the linear regression model, compared to using 6MWD alone as the predictive parameter.

## Materials and methods

This was a cross-sectional study approved by the Institutional Review Board of Shenzhen Second People's Hospital (Ethics approval number: KS20191119004). The study protocol

(Register number: ChiCTR1900028393) is available at the Chinese Clinical Trial Register Center website: www.chictr.org.cn. The protocol of this study is also available at: dx.doi.org/10.17504/protocols.io.6qpvr6pp2vmk/v1.

## Participants

People diagnosed with stroke and receiving treatment at the Second People's Hospital, Shenzhen, China, between December 2019 and December 2021, were invited to participate in the study through in-hospital poster advertising. Interested participants were screened for inclusion eligibility (see criteria below). The nature of the study was explained and written informed consent was obtained from all participants prior to data collection.

The inclusion criteria were: (1) age $\geq$ 18 years, (2) clinically diagnosed with ischemic and/or hemorrhagic stroke, (3) period since stroke diagnosis ranging from 3 to 12 months after diagnosis, (4) able to independently ambulate with or without an assistive device for $\geq$ 100 meters, (5) medically stable and with no significant limitation due to pain, and (6) able to clearly comprehend the exercise testing instructions.

The exclusion criteria were: (1) current use of beta-blocker medications; (2) neurological condition other than stroke and/or an orthopedic condition causing motor deficit (e.g. fracture, degenerative joint change, clinical instability of the hip or knee joint), (3) psychiatric impairment, such as severe depression or panic disorder, (4) pregnancy, (5) uncontrolled hypertension, arrhythmia, or an unstable cardiovascular status, (6) recent pulmonary embolism, subacute systemic illness or infection, and (7) brain injury affecting the respiratory and circulatory centers, e.g. brainstem injury.

## Procedure

Demographic data for each eligible participant (including age, sex, height, weight, body mass index (BMI), percent body fat and lean body mass) were recorded. The stroke type (cerebral infarction, intracerebral hemorrhage), medical history of comorbidity including hypertension, diabetes mellitus, cardiovascular disease, lipidemia, kidney disease, pulmonary disease, National Institutes of Health Stroke Scale (NIHSS), Modified Rivermead Mobility Index (MRMI), Berg Balance Scale (BBS), Barthel index (BI) were retrieved from the patient's medical record.

NHISS—a 15-item impairment scale used to measure stroke severity by means of a score ranging from 0 to 42 [18]. The higher the NHISS score, the more severe the stroke. MRMI is an 8-item scale with score ranging from 0 to 40 [19]. Higher MRMI scores imply a greater level of independence during transfers, balance and walking.

BBS is a scale which includes 14 items that measure balance with a total score between 0 and 56. A higher BBS score indicates better balance [20].

BI is a 10-item instrument assessing the basic activities of daily living for an individual. The minimum score of 0 reflects totally dependent, and a maximum score of 100 indicates totally independent living [21].

Participants were requested to attend the hospital cardiopulmonary laboratory twice, 72 hours apart. They were randomized to either a CPET or two 6MWTs at their first visit. The two 6MWTs were conducted consecutively with a separation of 30 minutes. Thus, if a patient had CPET during their first visit, the 6MWTs would be conducted 72 hours later, and vice versa. Each visit was scheduled at least 2 hours after a light meal. Participants were also requested to avoid caffeine-containing products, nicotine, and alcohol for at least 12 hours before attending the laboratory. Breath-by-breath oxygen consumption was recorded during

the CPET and cardiodynamic parameters (HR/SV/CO) were recorded with impedance cardiography (ICG) during both the CPET and 6MWTs (see below).

## Measurement of cardiac parameters

The HR, SV and CO during the tests were measured at 1-second intervals using a Physio-Flow®PF07 Enduro™ (PhysioFlow Enduro, Paris, France). The PhysioFlow®PF07 is a portable, non-invasive device that adopts real-time wireless monitoring of morphology-based impedance cardiography signals via a blue tooth USB adapter. Electrode placement was conducted as described by Tonelli and colleagues [22]. Auto-calibration of the device was performed as instructed by the manufacturer before data collection. Variables were measured by the Enduro™ at 1-second intervals, prior to, during and after, each 6MWT and CPET. HR was derived from the electrocardiograph (ECG). Variations in the impedance signal during cardiac ejection generate a specific waveform from which the SV was calculated [23]. CO was computed by multiplication of the SV and HR. The reliability of ICG in people after stroke has been previously reported [24].

## 6-minute walk test (6MWT)

The 6MWT was performed in a 30-meter hospital hallway. Two 6MWT trials were performed to accommodate any possible learning effect and to ensure maximal effort, each test was conducted according to the American Thoracic Society (ATS) guidelines [9]. A rest period of at least 30 minutes was allowed between tests. Each participant was asked to rest in a sitting position for 10 minutes before and after each 6MWT. Apart from ICG recording as described above, oxygen saturation (SpO$_2$) was recorded by a pulse oximeter (Heal Force, POD-3, China) immediately before, and at minute intervals during, and at the end of the 6MWT. Blood pressure (BP) was measured by an electronic blood pressure monitor (OMRON, U30, China) immediately before and after the 6MWT, and at 2-minute intervals during the 10-minute rest period post 6MWT. Perceived fatigue sensation (modified Borg 0–10 Scale [25]) was recorded immediately at the end of 6MWT. Both the 6-minute walk distance (6MWD) as well as the cardiodynamic data recorded during the 'better' performed 6MWT were used for analysis.

## Cardiopulmonary exercise test (CPET)

Participants were requested to perform a symptom limited, graded CPET on a cycle ergometer (Ergoline GmbH, ergoselect 200, Germany). Throughout the test, 12-lead electrocardiography was continuously recorded. Participants were required to wear a mask and breathe through a calibrated volume sensor attached to a metabolic cart (MasterScreen™ CPX, CareFusion, Germany). Breath-by-breath oxygen consumption, carbon dioxide consumption and respiratory exchange ratio (RER) were measured. The gas analysis system was fully calibrated immediately before every test in accordance with the manufacturer's instructions. HR, SV, and CO data were recorded by ICG at 1-second intervals during the CPET. Blood pressure was measured at 2-minute intervals, and SpO$_2$ was measured at 1-minute intervals during the CPET.

## CPET protocol

The CPET protocol commenced with a rest period of 3 minutes sitting on the cycle ergometer to establish a steady state, then a 3-minute warm-up stage with pedaling without resistance. Participants were then required to pedal at increasing intensity of 4–8 W increments each minute, to ensure that the total exercise time remained in a range of 8–12 minutes. Participants

were instructed to maintain a cycling speed of 55–65 revolutions per min. Strong verbal encouragement was given throughout the test. The test was terminated once the participant was unable to maintain the required pedaling rate despite encouragement, or should any signs of risk to health, as prescribed in the guidelines of the American College of Sports Medicine (ACSM) [26] become manifest. Respiratory exchange ratio (RER) > 1.15 and Borg scores at the level "very hard" were used to signify a "maximal" exercise test performance [27]. Participants were asked to rate their sensation of fatigue level immediately after the test using the modified Borg 0–10 scale administered during the 6MWT.

## Data analysis

All data were analyzed using IBM SPSS Statistics for Windows, Version 25.0 (Armonk, NY: IBM Corp). Demographic data and clinical characteristics for all participants were summarized using descriptive statistics. Variables of interval-ratio data meeting a normality assumption were compared using the independent t test for gender difference. The Chi-square test was used for the analysis of gender specific categorical data. HR, SV, CO were averaged every 10 seconds. Differences in HR, SV, CO recorded at rest, at the peak of CPET and at the end of the 6MWT were analyzed using repeated-measures ANOVA. The associations between measured VO$_{2peak}$ and recorded variables (including 6MWD, HR, SV, CO) at the end of 6MWT and at the end of the CPET were analyzed by Pearson's correlation coefficients. The p-value of <0.05 was considered statistically significant.

Similar statistical analyses, as described in our previous work, were adopted [17]. Three sets of multiple linear regression analyses were primarily performed to explore the optimal predictor variables of VO$_{2peak}$. The first regression equation was built by 6MWD as the single variable predicting VO$_{2peak}$. The second regression equation was generated using BMI, age, gender, time after stroke, 6MWD, plus HR, SV and CO recorded by ICG at the end of the better performed 6MWT as predictors for VO$_{2peak}$. The third regression equation utilised the same variables as in the 2$^{nd}$ equation but with peak HR, SV and CO measured during CPET. The stepwise backward regression method was adopted to determine the significant predictor variables to be retained in the regression equations. The appropriateness and precision of the regression parameters were evaluated with the squared multiple correlation ($R^2$), the standard error of estimate (SEE) as well as the 'SEE/meanVO$_{2peak}$' ratio, which was expressed as a percentage (SEE%). The predicted residual sum of squares (PRESS) statistic [28] was computed to estimate the degree of $R^2$ shrinkage ($R_p^2$) when the VO$_{2peak}$ regression equation was used for cross-validation across similar but independent samples. PRESS-derived $R_p^2$, SEE$_p$ and SEE$_p$% for each regression model were compared.

Two further analyses were conducted to examine the unique contribution of changes in HR or SV to any changes in CO, during 6MWT and CPET; forced entry regression method with HR change and SV change as predictor variables were used.

## Sample size estimation

The PASS 15.0.5 (Kaysville, Utah: NCSS) was used to calculate the sample size required for the multiple regression analyses. A sample size of at least 54 participants were required, with 8 predictor variables (age, gender, BMI, time after stroke, 6MWD, HR, SV, CO) modeled for estimated effect size of 0.30, α level of 0.05 and power of 0.80.

## Results

A total of 67 patients met the inclusion criteria and participated in the exercise tests. Eight participants either had RER below1.15 or were unable to tolerate the CPET and thus were

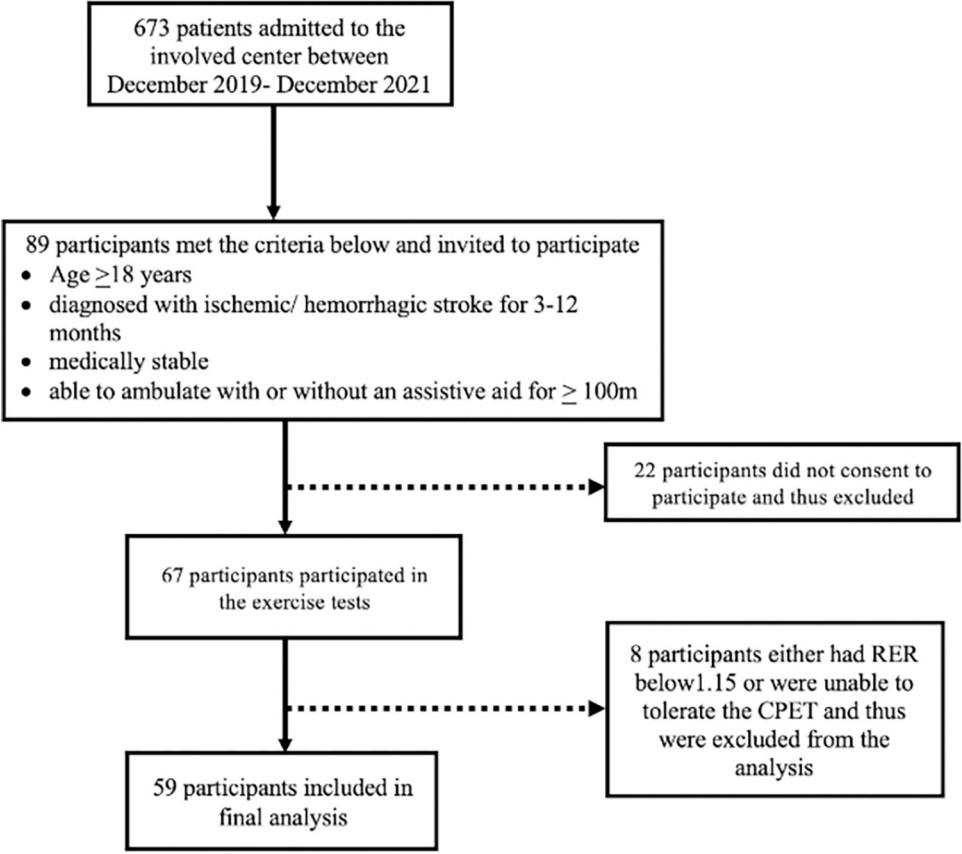

**Fig 1. A flow chart illustrating the number of patients entered for final analysis.**

excluded from the analysis. A flow chart illustrating the number of patients entered for final analysis is displayed in Fig 1.

The mean age of the 59 participants (52 males) included was 50.0±11.7 years. Demographic data and clinical characteristics of these participants are displayed in Table 1.

The 6MWD, peak HR, SV, CO during 6MWT and CPET and VO$_{2peak}$ recorded at CPET are displayed in Table 2. Peak HR and CO during CPET were significantly higher than that achieved at the end of the 6MWT, with a mean difference in HR and CO being 26 (95% CI: 22.3 to 29.3) and 2.3 (95% CI: 1.8 to 2.8), respectively. However, there was no difference in maximum SV between the two tests, the mean difference of maximum SV was 0.3 (95% CI: -3.2 to 2.5).

The VO$_{2peak}$ achieved during CPET correlated well with 6MWD (r = 0.66), peak HR (r = 0.82) and peak CO (r = 0.74) during CPET, as well as with HR (r = 0.73) and CO (r = 0.6) recorded at the end of the 6MWT. However, correlations between measured VO$_{2peak}$ and SV during either CPET (r = 0.39) and 6MWT (r = 0.4) were poor (Table 3).

## Regression analyses for prediction of VO$_{2peak}$

Three multiple linear regression models were generated for prediction of VO$_{2peak}$ (Table 4). Model 1, with 6MWD as the only predictor variable, produced a $R^2$ of 0.44 and a SEE% of 12.5%. Model 2 used SV and CO recorded during the maximal CPET as predictor variables and this equation produced a $R^2$ of 0.69 and SEE% of 9.2%. Model 3: 16.855+(-0.060 x age (years)) + (0.196 x BMI (kg/m$^2$) + (0.01 x 6MWD(m)) + (-0.416 x SV$_{6MWT}$ (ml)) + (3.587 x

**Table 1. Demographic data and clinical characteristics of the 59 participants.** Data in n (%) or mean±SD.

| | Total (n = 59) | Male (n = 52) | Female (n = 7) | P value |
|---|---|---|---|---|
| Age (years) | 50.0±11.7 | 49.6±11.6 | 53.1±12.5 | 0.451 |
| Height (cm) | 167 ± 6 | 168.2±4.9 | 159.0±3.9 | <0.001 |
| Weight (kg) | 65.5±8.6 | 66.5±8.2 | 58.5±9.0 | 0.020 |
| BMI(kg/m$^2$) | 23.4±2.7 | 23.4±2.5 | 23.3±4.1 | 0.915 |
| Lean body mass (kg) | 49.8±5.2 | 50.8±4.6 | 42.9±4.6 | <0.001 |
| Diagnosis | | | | |
| Cerebral hemorrhage | 21 (36%) | 17(33%) | 4 (57%) | 0.024 |
| Cerebral infarction | 38 (64%) | 35(67%) | 3 (43%) | |
| Lesion side | | | | |
| Left | 26(44%) | 22(42%) | 4(42%) | 0.393 |
| Right | 33(56%) | 30(58%) | 3(58%) | |
| Time after stroke (months) | | | | |
| <3 | 12 (23%) | 9(17%) | 3(43%) | 0.030 |
| 3–6 | 29 (49%) | 25(48%) | 4(57%) | |
| 7–12 | 18 (28%) | 18(35%) | 0(0%) | |
| Comorbidities | | | | |
| Hypertension | 24 (41%) | 22(43%) | 2(29%) | 0.024 |
| Diabetes mellitus | 28 (48%) | 24(46%) | 4(57%) | |
| Hyperlipidemia | 9 (15%) | 6(12%) | 3(43%) | |
| Heart disease | 1 (2%) | 1(2%) | 0(0%) | |
| Kidney disease | 0 (0%) | 0 (0%) | 0 (0%) | |
| Pulmonary disease | 0 (0%) | 0 (0%) | 0 (0%) | |
| Ambulation | | | | |
| Independent | 56 (97%) | 50(96%) | 6(86%) | 0.229 |
| With one walking stick | 3 (3%) | 2(4%) | 1(14%) | |
| NHISS score | 2.5±1.8 | 2.5±1.8 | 2.6±1.7 | 0.923 |
| MRMI score | 37.6±3.2 | 37.7±3.2 | 36.9±3.4 | 0.533 |
| BI score | 89.4±11.9 | 89.4±12.1 | 89.3±11.0 | 0.977 |
| Berg score | 49.8±9.3 | 49.9±9.7 | 48.6±5.8 | 0.767 |
| Borg score at the end of the test with greater 6MWD | 5±2 | 5±2 | 5±1 | 0.725 |
| Borg score at the end of the CPET | 8±1 | 8±1 | 8±1 | 0.775 |
| Peak work rate achieved during CPET (W) | 68.8±23.1 | 71.5±22.7 | 48.9±16.1 | 0.013 |
| VO$_{2peak}$ achieved during CPET (ml/min/kg) | 19.2±3.2 | 19.5±3.1 | 16.8±2.5 | 0.038 |
| 6MWD (m) | 294±131 | 298±131 | 265±135 | 0.539 |

BMI = Body Mass Index; NHISS = National Institutes of Health Stroke Scale; MRMI = Modified Rivermead Mobility Index; BI = Barthel Index; 6MWD = the farther distance covered in the two 6 minute walk tests; VO$_{2peak}$ = peak oxygen consumption, CPET = cardiopulmonary exercise test.

CO$_{6MWT}$ (L/min)) was associated with the highest $R^2$ (0.7) and lowest SEE (1.75 mL/kg/min) and SEE% (9.1); this model is comparable to Model 2 for prediction of VO$_{2peak}$ measured during CPET from data measured during the 6MWT. The PRESS derived regression model illustrated a $R_p^2$ of 0.66, SEE$_p$ of 1.83mL/kg/min and SEE p % of 9.5%. This model also produced the highest correlation coefficient between measured and predicted VO$_{2peak}$ (r = 0.847, p<0.001) (Table 5).

## Regression analyses to determine unit contribution of HR or SV changes to CO

Multiple linear regression analyses further revealed that the changes in HR or SV contributed equally to the change of CO during the 6MWT. During the CPET however, the change in CO was mainly a response to HR change rather than changes in SV (Table 6).

**Table 2. Cardiodynamic parameters measured by ICG at end of 6MWT and CPET.** Data in mean±SD.

| | 6MWT | | | | | | | CPET | | | | | | |
|---|---|---|---|---|---|---|---|---|---|---|---|---|---|---|
| | | Heart rate | | Stroke volume | | Cardiac output | | VO$_2$ | Heart rate | | Stroke volume | | Cardiac output | |
| | 6MWD (m) | RHR-6MWT (bpm) | PHR-6MWT (bpm) | RSV-6MWT (ml) | PSV-6MWT (ml) | RCO-6MWT (L/min) | PCO-6MWT (L/min) | Peak VO$_2$ (ml/min/kg) | RHR-CPET (bpm) | PHR-CPET (bpm) | RSV-CPET (ml) | PSV-CPET (ml) | RCO-CPET (l/min) | PCO-CPET (l/min) |
| Male n = 52 | 298 ±131 | 81±11 | 110±5 | 66.7 ±10.8 | 86.6±8.8 | 5.3±0.8 | 9.5±1.2 | 19.5±3.1 | 82±11 | 136±14 | 66.6 ±11.6 | 87.3 ±8.9 | 5.4±0.8 | 11.9±2 |
| Female n = 7 | 265 ±135 | 80±8 | 104±5 | 60.7 ±7.4 | 84.4±9.6 | 4.8±0.6 | 8.8±1.4 | 16.8±2.5 | 85±10 | 125±13 | 63±10 | 82.1 ±9.1 | 5.4±1.1 | 10.3±1.8 |
| Total n = 59 | 294 ±131 | 81±11 | 109±6 | 66 ±10.6 | 86.3±8.8 | 5.3±0.8 | 9.4±1.2 | 19.2±3.2 | 82±11 | 135±14 | 66.2 ±11.4 | 86.6±9 | 5.4±0.8 | 11.7±2 |

ICG = impedance cardiography; 6MWT = 6 minute walk test; 6MWD = the farther distance covered in the two 6MWTs; CPET = cardiopulmonary exercise test; RHR$_{-6MWT}$ = Resting heart rate–6MWT; PHR$_{-6MWT}$ = Peak heart rate–6MWT; RSV$_{-6MWT}$ = Resting stroke volume–6MWT; PSV$_{-6MWT}$ = Peak stroke volume–6MWT; RCO$_{-6MWT}$ = Resting cardiac output–6MWT; PCO$_{-6MWT}$ = Peak cardiac output–6MWT; VO$_2$ = oxygen consumption; RHR$_{-CPET}$ = Resting heart rate–cardiopulmonary exercise test; PHR$_{-CPET}$ = Peak heart rate–cardiopulmonary exercise test; RSV$_{-CPET}$ = Resting stroke volume–cardiopulmonary exercise test; PSV$_{-CPET}$ = Peak stroke volume–cardiopulmonary exercise test; RCO$_{-CPET}$ = Resting cardiac output–cardiopulmonary exercise test; PCO$_{-CPET}$ = Peak cardiac output–cardiopulmonary exercise test.

## Discussion

This study explored the hemodynamic responses recorded during a CPET and a 6MWT in the same cohort of people after stroke. Our study revealed that measured VO$_{2peak}$ during the CPET correlated well with HR, SV and CO data recorded during both CPET and the 6MWT (Table 4). The 6MWT is a convenient field test commonly used as an alternative method of assessment of aerobic fitness, especially in people with chronic impairment of cardiorespiratory and neurological function [11,29]; and as the 6MWT is a sub-maximal exercise test, it is not surprising that the peak heart rate and maximal cardiac output achieved at the end of the 6MWT were lower than peak HR and CO recorded during the CPET (Table 2). Consequently, the level of exertion as expressed by the Borg's score recorded at the end of the 6MWT was only 5 compared to 8 at the end of the CPET, suggesting that the 6MWT appeared less strenuous compared to the exercise demands during the CPET.

**Table 3. Correlation between VO$_{2peak}$ and 6MWD, maximal HR, SV and CO during both 6MWT and CPET.**

| Variable | Correlation with VO$_{2peak}$ Pearson r | P value |
|---|---|---|
| HR$_{Peak}$ (bpm) | 0.82 | <0.001 |
| SV$_{Peak}$ (ml) | 0.39 | <0.05 |
| CO$_{Peak}$ (L/min) | 0.74 | <0.001 |
| HR$_{end}$ (bpm) | 0.73 | <0.001 |
| SV$_{end}$ (ml) | 0.40 | <0.05 |
| CO$_{end}$ (L/min) | 0.60 | <0.001 |
| 6MWD | 0.66 | <0.001 |

6MWT = 6–minute walk test; 6MWD = the farther distance covered in the two 6MWTs; CPET = Cardiopulmonary exercise testing; VO$_{2peak}$ = measured peak oxygen consumption at CPET; HR$_{Peak}$ = peak heart rate at CPET; SV$_{Peak}$ = peak stroke volume at CPET; CO$_{Peak}$ = peak cardiac output at CPET; HR$_{end}$ = heart rate at the end of 6MWT; SV$_{end}$ = stroke volume at the end of 6MWT; CO$_{end}$ = cardiac output at the end of 6MWT.

**Table 4. Multiple regression analyses for prediction of $VO_{2peak}$.**

| Model | Predictor variables ($X_n$) | | | Coefficients | β | $R^2$ | SEE | SEE % | $R_p^2$ | $SEE_p$ | $SEE_p$% | Regression equation |
|---|---|---|---|---|---|---|---|---|---|---|---|---|
| | Variables put into model | Removed from model because of non-significance | Retained | | | | | | | | | |
| Model 1 6MWD as sole predictor | 6MWD | - | constant 6MWD | 14.442 0.016 | 0.663 | 0.44 | 2.4 | 12.5 | 0.39 | 2.46 | 12.8 | $VO_{2peak} = 14.442 + (0.016 \times 6MWD)$ |
| Model 2 with data from CPET | BMI, Age, Gender, Time after stroke, $HR_{peak}$, $SV_{peak}$, $CO_{peak}$ | Age Gender BMI $HR_{peak}$ | constant $SV_{peak}$ $CO_{peak}$ | 15.869 -0.242 2.076 | -0.685 1.303 | 0.69 | 1.77 | 9.2 | 0.67 | 1.82 | 9.5 | $VO_{2peak} = 15.869 + (-0.242 \times SV_{peak})$ $+(2.076 \times CO_{peak})$ |
| Model 3 with data from 6MWT | BMI, Age, Gender, Time after stroke, 6MWD, $HR_{end}$, $SV_{end}$, $CO_{end}$ | Gender Time after stroke $HR_{end}$ | constant BMI Age 6MWD $SV_{end}$ $CO_{end}$ | 16.855 0.196 -0.060 0.010 -0.416 3.587 | 0.167 -0.223 0.398 -1.158 1.402 | 0.70 | 1.75 | 9.1 | 0.66 | 1.83 | 9.5 | VO2peak = 16.855+(-0.060 x age) + (0.196 x BMI)+ (0.01 x 6MWD)+ (-0.416 x $SV_{end}$) +(3.587 x $CO_{end}$) |
| Model 3a data from 6MWT excluding age, BMI | $HR_{end}$, $SV_{end}$, $CO_{end}$ 6MWD | $HR_{end}$ | constant 6MWD $SV_{end}$ $CO_{end}$ | 15.253 0.009 -0.364 3.465 | 0.383 -1.015 1.355 | 0.64 | 1.91 | 9.9 | 0.61 | 1.98 | 10.3 | $VO_{2peak} = 15.253 + (0.009 \times 6MWD)$ + (-0.364 x $SV_{end}$) + (3.465 x $CO_{end}$) |

$VO_{2peak}$ = Peak oxygen consumption during the CPET; 6MWD = the farther distance covered in the two 6minute walk tests; $HR_{end}$ = Heart rate at the end of 6MWT; $SV_{end}$ = Stroke volume at the end of 6MWT; $CO_{end}$ = Cardiac output at the end of 6MWT; $HR_{peak}$ = Peak heart rate during the CPET; $SV_{peak}$ = Peak stroke volume during the CPET; $CO_{peak}$ = Peak cardiac output during the CPET.

Poor cardiovascular fitness in people after stroke affects quality of life [3]. Exercises that aim to increase aerobic fitness are recommended for inclusion in rehabilitation programs for stroke survivors [6,8]. An accurate assessment of $VO_{2peak}$ is necessary to prescribe an optimum exercise program which permits accurate monitoring and evaluation, of aerobic capacity. The primary aim of this current study was to identify an equation that most accurately predicts $VO_{2peak}$ in people after stroke. Cardiodynamic data recorded from the 6MWT and CPET were used to generate an equation to predict the $VO_{2peak}$ achieved during the CPET, the gold

**Table 5. Correlation between measured and predicted $VO_{2peak}$ generated by different equation models.**

| $VO_{2peak}$ prediction Model | | Correlation with measured $VO_{2peak}$ r values | P value |
|---|---|---|---|
| Model 1 | $VO_{2peak}$ = 14.442 + (0.016 x 6MWD) | 0.663 | <0.001 |
| Model 2 | $VO_{2peak}$ = 15.869 + (-0.242x $SV_{peak}$) +(2.076x$CO_{peak}$) | 0.838 | <0.001 |
| Model 3 | $VO_{2peak}$ = 16.855+(-0.060 x age) + (0.196 x BMI) + (0.01 x 6MWD) + (-0.416 x $SV_{end}$) +(3.587 x $CO_{end}$) | 0.847 | <0.001 |

r = Pearson correlation coefficient.

$VO_{2peak}$ = Peak oxygen consumption during the CPET; 6MWD = the farther distance covered in the two 6minute walk tests; $SV_{end}$ = Stroke volume at the end of 6MWT; $CO_{end}$ = Cardiac output at the end of 6MWT; $SV_{peak}$ = Peak stroke volume during the CPET; $CO_{peak}$ = Peak cardiac output during the CPET.

**Table 6. Contribution of unique changes in HR and SV to changes in CO (Multiple linear regression analyses using change in CO as an outcome variable).**

| | At the end of 6MWT | | | At the end of CPET | | |
|---|---|---|---|---|---|---|
| | **male** | **Female** | **all** | **male** | **female** | **All** |
| HR change standardized beta coefficient | 0.73 | 0.64 | 0.73 | 0.79 | 0.71 | 0.78 |
| SV change Standardized beta coefficient | 0.69 | 0.44 | 0.67 | 0.55 | 0.34 | 0.52 |
| HR change semipartial correlation | 0.73 | 0.49 | 0.73 | 0.78 | 0.46 | 0.78 |
| SV change semipartial correlation | 0.69 | 0.34 | 0.67 | 0.55 | 0.21 | 0.52 |
| HR change unique contribution to CO change (%) | 54 | 24 | 53 | 61 | 21 | 60 |
| SV change unique contribution to CO change (%) | 48 | 11 | 45 | 30 | 5 | 27 |
| 6MWD (m) | 298±131 | 265±135 | 294±131 | - | - | - |
| Peak VO$_2$ (ml/min/kg) | - | - | - | 19.5±3.1 | 16.8±2.5 | 19.2±3.2 |

6MWT = 6–minute walk test; CPET = Cardiopulmonary exercise testing; 6MWD = the farther distance covered in the two 6MWTs; HR = heart rate; SV = stroke volume; CO = cardiac output.

standard for evaluation of maximal aerobic capacity. In accord with our previous work with young healthy adults [17], this study reveals that the inclusion of age, BMI and SV and CO recorded by ICG (a non-invasive and simple to apply procedure during a 6MWT) generated more accurate prediction equation for VO$_{2peak}$ achieved by the CPET in our cohort of patients after stroke. Using 6MWD as the sole predictor of VO$_{2peak}$, the correlation coefficient between predicted and measured VO$_{2peak}$ was only 0.66, but this correlation coefficient increased to 0.85 with the inclusion of age, BMI, and SV and CO recorded during 6MWT (Table 5). Not only was this model associated with a higher $R^2$ and lower SEE and SEE% compared to the equation which relied on 6MWD as the predictor of VO$_{2peak}$, the $R^2$ and SEE values associated with this equation suggest that the predictive power of this equation is as accurate as using an individual's peak SV and CO during the gold standard CPET as predictors of VO$_{2peak}$ (Model 2, Table 4). Additional analysis showed that with removal of the predictors age and BMI (Model 3a, Table 4), the $R^2$ value of the equation remains comparable to that of Model 2. This suggests that SV and CO measured by ICG during a 6MWT are strong predictors of the measured VO$_{2peak}$ in our cohort of participants with stroke. The $R^2$ and SEE values of VO$_{2peak}$ prediction equation in people with aneurysmal subarachnoid hemorrhage was previously reported by Harmsen and colleagues to be 0.56 and 4.12 respectively [30]. The mean age of our patient cohort and that reported by Harmsen's group were similar, but the mean 6MWD achieved by our cohort (294 ±131m) was comparatively much lower than the Harmsen cohort (498±98m). The 6MWD has been commonly used to predict VO$_{2peak}$ but because functional capacity varies according to stroke severity, 6MWD has not been considered as a suitable indicator of aerobic capacity in people after stroke [31]. This current study shows that inclusion of SV and CO data recorded during a 6MWT provided a more precise prediction of maximal aerobic capacity in our stroke cohort.

It is interesting to note that while the CO during CPET was significantly higher than that recorded during a 6MWT, the maximal SV recorded during both tests were similar (Tables 2), inferring that the extra CO demands during CPET were met by way of an increase in HR, rather than SV. This inference is further supported by analysis of the standardized beta

coefficient associated with changes in HR and SV. As illustrated in Table 6, the unique contribution of HR and SV changes to the change in CO during the 6MWT was 53% and 45% respectively. However, the contribution of HR change to CO was twice the change in SV seen during the CPET, affirming that the increase in HR was the main contributor to the increase in CO during the CPET. This finding accords with our previous report [17]. There are multiple explanations for dampened SV responses to increased exercise demands in the stroke population. It has been shown in athletes with paraplegia, that an increase in CO was accompanied by a significant increase in HR [32], whereas able-bodied individuals attained CO with a lower HR and higher SV; these authors postulated that this anomaly was due to a comparative reduction in venous return during exercise in people with paraplegia, and consequently a reduced SV. Further, it has been postulated that poor functional capacity is associated with suboptimal left ventricular diastolic function in people after stroke [2], it thereby follows that impaired left ventricular diastolic function could dampen the SV response to increased exercise intensity in this population. Similarly, in people with heart failure, maintenance of CO during CPET was reportedly associated with a compensatory increase in HR [33]; an increase in heart rate leads to a reduced diastolic filling time thereby reducing stroke volume [34].

### Implications of the study

Although the 6-minute walk test (6MWT) is a common field test used to monitor and evaluate submaximal aerobic capacity, the predictive power of 6MWD on VO$_{2peak}$ in people after stroke has been considered unreliable. The 6MWD is an outcome often used to reflect the VO$_{2peak}$ (aerobic capacity) of the participant, however, in the stroke population, the 'distance' covered during the test can be influenced by non-haemodynamic factors. Heart rate (HR), stroke volume (SV) and cardiac output (CO) are primary contributing factors to exercise capacity and can now be measured noninvasively and conveniently by ICG during a 6MWT. Information from these 'direct' cardiodynamic factors should more reliably inform the clinician of the aerobic capacity of the participant. Further, the predictive power of 6MWD for VO$_{2peak}$ in the stroke population has been questioned [10]. Our findings show that the inclusion of SV and CO during 6MWT combined with 6MWD improves the predictive power of aerobic capacity in people after stroke compared to the predictive power of 6MWD alone. Thus the accuracy of a prescription for an effective rehabilitation exercise program can be enhanced. However, whether using this new predictive equation results in a more effective outcome following the prescribed program warrants further investigation.

### Limitations of the study

The majority (90%) of patients admitted to our center during the data collection period were males. Unfortunately, it is not clear why this occurred. The participants were those with moderate motor deficits post-stroke and a mean 6MWD of less than 300m. It may not be appropriate to apply inferences drawn from our data analysis to female patients, however the pattern of response to exercise stress between male and female patients appeared to be similar. Our data suggest that further studies with a larger sample size to explore any effect of age, gender, and a wide range of motor capacity on ICG measured cardiodynamic parameters in response to exercise training, are warranted.

### Conclusion

This study demonstrated that inclusion of SV and CO measured during the 6MWT in stroke patients further improved the VO$_{2peak}$ prediction power compared to using 6MWD as a lone

predictor. Further studies exploring the regression model with wider demographic profiles are warranted.

## Supporting information

**S1 Checklist. STROBE statement—checklist of items that should be included in reports of observational studies.**
(DOCX)

## Author Contributions

**Conceptualization:** Fang Liu, Alice Y. M. Jones, Yulong Wang.

**Data curation:** Fang Liu, Raymond C. C. Tsang, Fubing Zha, Mingchao Zhou, Kaiwen Xue, Zeyu Zhang.

**Formal analysis:** Fang Liu, Alice Y. M. Jones, Raymond C. C. Tsang.

**Funding acquisition:** Yulong Wang.

**Investigation:** Fang Liu, Fubing Zha, Mingchao Zhou, Kaiwen Xue, Zeyu Zhang.

**Methodology:** Fang Liu, Alice Y. M. Jones, Raymond C. C. Tsang, Fubing Zha.

**Project administration:** Yulong Wang.

**Software:** Fang Liu, Raymond C. C. Tsang.

**Supervision:** Alice Y. M. Jones, Yulong Wang.

**Validation:** Alice Y. M. Jones, Fubing Zha, Mingchao Zhou, Kaiwen Xue, Zeyu Zhang, Yulong Wang.

**Visualization:** Kaiwen Xue, Yulong Wang.

**Writing – original draft:** Fang Liu, Alice Y. M. Jones, Raymond C. C. Tsang.

**Writing – review & editing:** Fang Liu, Alice Y. M. Jones, Raymond C. C. Tsang.

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
