## [Decision Letter · Decision Letter 0]

12 Jul 2022

PONE-D-22-12074Stroke volume and cardiac output during 6 minute-walk tests are strong predictors of maximal oxygen uptake in people with strokePLOS ONE

Dear Dr. Jones,

Thank you for submitting your manuscript to PLOS ONE. After careful consideration, we feel that it has merit but does not fully meet PLOS ONE’s publication criteria as it currently stands. Therefore, we invite you to submit a revised version of the manuscript that addresses the points raised during the review process.

For this initial peer review cycle fortuitously three content experts have provided input. Whilst all expressed overall positive sentiments on the submission, numerous suggested amendments and queries have been offered for consideration, as detailed explicitly by each reviewer below.

We look forward to receiving your revised manuscript.

Kind regards,

Shane Patman, PhD

Academic Editor

PLOS ONE

Journal Requirements:

“This research was funded by the Sanming Project of Medicine in Shenzhen. Grant number SZSM201512011.”

Reviewers' comments:

Reviewer's Responses to Questions

**Comments to the Author**

1. Is the manuscript technically sound, and do the data support the conclusions?

Reviewer #1: Yes

Reviewer #2: Yes

Reviewer #3: Yes

2. Has the statistical analysis been performed appropriately and rigorously? 

Reviewer #1: Yes

Reviewer #2: I Don't Know

Reviewer #3: I Don't Know

3. Have the authors made all data underlying the findings in their manuscript fully available?

Reviewer #1: Yes

Reviewer #2: Yes

Reviewer #3: Yes

4. Is the manuscript presented in an intelligible fashion and written in standard English?

Reviewer #1: Yes

Reviewer #2: Yes

Reviewer #3: Yes

5. Review Comments to the Author

Reviewer #1: Dear Editors nad Authors

Thank you for the opportunity to review the paper "Stroke volume and cardiac output during 6 minute-walk tests are strong predictors of maximal oxygen uptake in people with stroke" that concerns interesting research area - how to simply and objectively assess exercise capacity in patients with stroke. The authors use measurements of cardiovascular hemodynamics while 6MWT to predict VO2 peak in CPET. The concept is good because CPET is quite a difficult and time comsuming procedure. The results of the study revealed that VO2 peak can be estimated from 6MWT+ICG with clinically acceptable accuracy. The paper is well written.

I have only minor comments to be considered:

1. Abstract verses 16-18 - please put units after numbers, not in brackets .ie. "6MWD was 294+/-13 m"

2. Please rephrase the second sentence of Introduction (verses 29-31) to be more general and simply, i.e. "Peak oxygen consumption (VO2 peak) measured after stroke is usually significantly lower than in sex-matched...."

3. Please provide info how the data was transferred from Physioflow device to the Physioflow software - was it real-time wireless transfer or did you used memory card with post-exam cable transfer to PC?

4. Please comment on possible low quality of ICG while CPET and 6MWT - were any problems with it. If yes - please comment in Limitations.

5. verse 106 - provide full name of 6MWT in subheading

6. Table 1 - "Duration of stroke" - better "Time from stroke"

7. Verse 237-240 - the difference between model 2 and 3 are very slight, do not write that model 2 was better, it was comparable (as you correctly write in discussion).

Reviewer #2: Overall, a thorough and well-executed work that is clear and well described.

However, I am missing a small important addition in the introduction in relation to why this study is relevant, which the authors argue for in the discussion.

I would also find it relevant to elaborate a bit more about how these findings is relevant and can contribute in the clinical practice of stroke rehabilitation.

If the authors will consider the following comments, I find the manuscript contributing to gaining knowledge in the field of rehabilitation stroke survivors.

Comments for the manuscript:

In the introduction (line 27-52) I lack to understand why it is important to ass ICG to 6MWT in stroke patients? Why is it not enough just to have the 6MWT as an expression for the submaximal VO2test?

How and why is this important in the clinic?

It is presented in the discussion line 306-307, but this should be presented in the introduction as well.

Methods:

Line 71-72: Did you use a score for the ambulation ability such as FAC?

Line 85-87 “National Institutes of Health Stroke Scale (NIHSS), 86 Modified Rivermead Mobility Index (MRMI), Berg Balance Scale (BBS), Barthel 87 index (BI)”

The tests should be short described according to score range indication no/severe dependence/impairment and a reference.

Line 88-90: “Participants were then requested to attend the hospital cardiopulmonary laboratory 89 twice (72 hours apart) to perform, in random order, a progressive cycle ergometer test 90 or two 6MWTs (at least 30 minutes apart)”.

I do not understand. Did the participants first do the one kind of test and then the day after the other kind of test? I am not sure by the description in this section.

Line 193-194” The VO2peak achieved during CPET correlated well with 6MWD, peak HR and peak 194 CO during CPET, as well as with HR and CO recorded at the end of the 6MWT”

You are stating something but not presenting the numbers. Please present the numbers and move the statement to the discussion section.

Line 221: Table 2 MWD is not clarified in the table note

Line 224: Table 3 Why is MWD represented as it is stated that the besto of the two tests is represented. What is it compared with? The same goes with COPeak (L/min).

Regression analyses for prediction of VO2peak line 233-245 I must say it I rather confusing but I am not the right person to verify if this method is the correct.

Discussion:

Results line 185: 59 participants (52 males)

Howcome som many males? Do you have any comment on that?

Line 188-191:

“Peak HR and CO 189 during CPET were significantly higher than that achieved at the end of the 6MWT, with 190 a mean difference in HR and CO being 26 (95% CI: 22.3 to 29.3) and 2.3 (95% CI: 1.8 191 to 2.8), respectively.”

That is some mean differences, why?

Your findings: SV and CO measured during the 6MWT in stroke patients further improved the VO2peak prediction power compared to using 6MWD as a lone predictor.

–By how much is the prediction improved in including impedance cardiography (ICG) during a 6MWT?? And is the difference clinical relevant?

I would find it relevant to elaborate a bit more about how these findings is relevant and can contribute in the clinical practice of stroke rehabilitation.

Reviewer #3: Dear Authors

I appreciate the authors and PLOS ONE for the opportunity to evaluate this paper. The authors led an interesting study that the prediction of VO2peak can be improved by the inclusion of cardiovascular indices derived by impedance cardiography (ICG) during the 6MWT in people with stroke. They found that the prediction equation with inclusion of cardio dynamic variables: 16.855 + (-0.060 x age) + (0.196 x BMI) +20 (0.01 x 6MWD) + (-0.416 x SV6MWT) + (3.587 x CO 6MWT) has a higher squared multiple correlation (R2) and a lower standard error of estimate (SEE) and SEE% compared to the equation using 6MWD as the only predictor. These findings will be of interest to clinicians, as well as researchers in the field.

I have following concerns.

Introduction

1. What is the clinical implication of this study? Is it significant that it has been conducted in young adults but not in stroke? If so, I would think a comparison with trends in young adults would be in order.

Methods

2. Are patients with arrhythmias excluded? There are descriptions of uncontrolled arrhythmias, but what about controlled Af? Additionally, were there any patients with brainstem lesions? There may be some impact because the respiratory and circulatory centers are located.

Results

3. A patient flow chart of inclusion/exclusion criteria may be helpful to understand the patient’s characteristics.

4. My greatest concern is that most of the subjects are male. It is mentioned in the limitation, but we need to be very careful about generalizing. Authors should consider presenting male-only results.

5. I think the p-value in Table 1 should be a specific number, not p>0.05.

6. Kidney disease and pulmonary disease were recorded, but there are no data in Table1.

7. Table 6 “At the end of end”. Is it “At the end of 6MWT”?

Discussion

8. It may not be appropriate to apply inferences drawn from our data analysis to female patients, however the pattern of response to exercise stress between male and female patients appeared to be similar.

The tendency of the pattern of response may be similar, but the degree to which they are different is quite different.

6. PLOS authors have the option to publish the peer review history of their article (what does this mean?). If published, this will include your full peer review and any attached files.

Reviewer #1: **Yes: **Paweł Krzesiński

Reviewer #2: No

Reviewer #3: No

---

## [Author Response · Author response to Decision Letter 0]

26 Jul 2022

We are grateful for all reviewers’ valuable and insightful comments, and we have revised our manuscript following their suggestions.

Reviewer #1: 

Dear Editors and Authors

Thank you for the opportunity to review the paper "Stroke volume and cardiac output during 6 minute-walk tests are strong predictors of maximal oxygen uptake in people with stroke" that concerns interesting research area - how to simply and objectively assess exercise capacity in patients with stroke. The authors use measurements of cardiovascular hemodynamics while 6MWT to predict VO2 peak in CPET. The concept is good because CPET is quite a difficult and time consuming procedure. The results of the study revealed that VO2 peak can be estimated from 6MWT+ICG with clinically acceptable accuracy. The paper is well written.

Authors response: We are grateful for Reviewer 1’s supportive comments.

I have only minor comments to be considered:

1. Abstract verses 16-18 - please put units after numbers, not in brackets .ie. "6MWD was 294+/-13 m"

Authors response: Units are now inserted as suggested. Please see lines 16-19 in the Abstract.

2. Please rephrase the second sentence of Introduction (verses 29-31) to be more general and simply, i.e. "Peak oxygen consumption (VO2 peak) measured after stroke is usually significantly lower than in sex-matched...."

Authors response: The sentence is modified as suggested. Please see lines 30-32.

3. Please provide information how the data was transferred from Physioflow device to the Physioflow software - was it real-time wireless transfer or did you used memory card with post-exam cable transfer to PC?

Authors response: Data from the Physioflow device was transmitted directly to the Physioflow software via real-time wireless transfer. This information is now included in the Methodology section, under Measurement of Cardiac parameters, lines 120-124. 

4. Please comment on possible low quality of ICG while CPET and 6MWT - were any problems with it. If yes - please comment in Limitations.

Authors response: We did not encounter any issues associated with the quality of ICG data recorded. Signals appeared stable during both CPET and 6MWT. 

5. verse 106 - provide full name of 6MWT in subheading.

Authors responses: The full name of 6MWT has been added in the subheading, line 132.

6. Table 1 - "Duration of stroke" - better "Time from stroke"

Authors response: “Duration of stroke” is now replaced with “Time after stroke” in the text and in Tables 1 and 4.

7. Verse 237-240 - the difference between model 2 and 3 are very slight, do not write that model 2 was better, it was comparable (as you correctly write in discussion).

Authors response: This sentence is now reworded as suggested, please see lines 272-273.

Reviewer #2: 

Overall, a thorough and well-executed work that is clear and well described.

However, I am missing a small important addition in the introduction in relation to why this study is relevant, which the authors argue for in the discussion.

I would also find it relevant to elaborate a bit more about how these findings is relevant and can contribute in the clinical practice of stroke rehabilitation.

If the authors will consider the following comments, I find the manuscript contributing to gaining knowledge in the field of rehabilitation stroke survivors.

Authors response: We are thankful for Reviewer 2’s insightful comments, we have included a more detailed explanation of the rationale for our study in the Introduction, as suggested.

Comments for the manuscript:

1. In the introduction (line 27-52) I lack to understand why it is important to assess ICG to 6MWT in stroke patients? Why is it not enough just to have the 6MWT as an expression for the submaximal VO2test?

Authors response: The primary aim of our study was indeed an attempt to answer this important question. The outcome used in a 6MWT is the distance covered during this submaximal test and the distance is used to reflect the VO2peak (aerobic capacity) of the participant. However, the ‘distance’ covered during the test in the stroke population can be influenced by non-haemodynamic factors. Heart rate, stroke volume and cardiac output are the primary contributing factors to exercise capacity. As these can now be measured noninvasively and conveniently during a 6MWT, we believe these ‘direct’ factors measured during the stress of an exercise test could more reliability reflect the aerobic capacity of the participant, than merely the distance. Further, the predictive power of 6MWD for VO2peak in the stroke population has been questioned. This forms the basis of the rationale of this study. We have now included this explanation in the introduction. Please refer to lines 45-54.

2. How and why is this important in the clinic? It is presented in the discussion line 306-307, but this should be presented in the introduction as well.

Authors response: The importance of using an accurate prediction of aerobic capacity to guide exercise prescription is now included in the Introduction, please see lines 35-39.

Methods:

3. Line 71-72: Did you use a score for the ambulation ability such as FAC?

Authors response: Thank you for this suggestion. In hindsight, it would have been an informative addition to reflect the mobility status of our cohort. In this study, apart from the 6MWD, we have only included the Modified Rivermead Mobility Index (MRMI) to reflect the mobility status of our participants.

4. Line 85-87 “National Institutes of Health Stroke Scale (NIHSS), Modified Rivermead Mobility Index (MRMI), Berg Balance Scale (BBS), Barthel index (BI)”

The tests should be short described according to score range indication no/severe dependence/impairment and a reference.

Authors response: A short description of these instruments is now included. Please see lines 100-108.

5. Line 88-90: “Participants were then requested to attend the hospital cardiopulmonary laboratory 89 twice (72 hours apart) to perform, in random order, a progressive cycle ergometer test 90 or two 6MWTs (at least 30 minutes apart)”. —I do not understand. Did the participants first do the one kind of test and then the day after the other kind of test? I am not sure by the description in this section.

Authors response: We apologise for the confusion. Description of this procedure is now clarified. Please see lines 109-113. 

6. Line 193-194” The VO2peak achieved during CPET correlated well with 6MWD, peak HR and peak CO during CPET, as well as with HR and CO recorded at the end of the 6MWT”—You are stating something but not presenting the numbers. Please present the numbers and move the statement to the discussion section.

Authors response: The correlation coefficients were now included in the text (lines 237-240) as well as in Table 3. 

7. Line 221: Table 2 6MWD is not clarified in the table note

Authors response: 6MWD is now defined in the table note. Please refer to the updated Table 2.

8. Line 224: Table 3 Why is 6MWD represented as it is stated that the best of the two tests is represented. What is it compared with? The same goes with COpeak (L/min).

Authors response: Two 6MWTs were performed by each participant, 30 min apart. We selected the better 6MWT, i.e. the test that resulted in the farthest 6MWD as the representation of the 6MWT. This is now clarified in the table note of Tables 2 and 3 as well as in the text. Please refer to lines 144-146.

9. Regression analyses for prediction of VO2peak line 233-245 I must say it I rather confusing but I am not the right person to verify if this method is the correct.

Authors response: Presentation of results of the multiple linear regression analyses was confirmed and checked by our statistician and is consistent with previous work published in PLoS One. We are happy to follow any suggestions which could make the interpretation easier.

Discussion:

10. Results line 185: 59 participants (52 males) —How come so many males? Do you have any comment on that?

Authors response: Indeed, this is a limitation of our study. For reasons unknown to us, the majority of patients admitted to our center who met the inclusion criteria were males during the data collection period. We were not able to explain this but have addressed this under the Limitations of the study. Please see line 405-410. 

11. Line 188-191: “Peak HR and CO during CPET were significantly higher than that achieved at the end of the 6MWT, with a mean difference in HR and CO being 26 (95% CI: 22.3 to 29.3) and 2.3 (95% CI: 1.8 191 to 2.8), respectively.”

—That is some mean differences, why?

Authors response: We postulate that the differences were a result of the fact that the 6MWT is only a submaximal exercise test while the CPET is a maximal exercise test. This observation is explained in the discussion section, please refer to lines 325-331.

12. Your findings: SV and CO measured during the 6MWT in stroke patients further improved the VO2peak prediction power compared to using 6MWD as a lone predictor.

–By how much is the prediction improved in including impedance cardiography (ICG) during a 6MWT?? And is the difference clinical relevant?

Authors response: In Model 1, the 6MWD was the only predictor variable and the results showed a small R2 of 0.44 and a larger SEE% of 12.5%. However, when the SV and CO were added with 6MWD, the R2 increased to 0.64 and the SEE% decreased to 9.9% (as shown in Table 5). A higher R2 value suggests a more reliable and stable predictive power, and a lower SEE% suggests a lower risk of error in estimation. Thus, mathematically this improvement is considered significant. However, the clinical relevance of this new predictive equation needs further investigation. We have included a paragraph on “Implications of the study” to address this important point. Please see lines 387-403.

13. I would find it relevant to elaborate a bit more about how these findings is relevant and can contribute in the clinical practice of stroke rehabilitation.

Authors response: We hope the paragraph “Implications of the study” mentioned above suitably addresses this point. 

Reviewer #3: 

Dear Authors

I appreciate the authors and PLOS ONE for the opportunity to evaluate this paper. The authors led an interesting study that the prediction of VO2peak can be improved by the inclusion of cardiovascular indices derived by impedance cardiography (ICG) during the 6MWT in people with stroke. They found that the prediction equation with inclusion of cardio dynamic variables: 16.855 + (-0.060 x age) + (0.196 x BMI) +20 (0.01 x 6MWD) + (-0.416 x SV6MWT) + (3.587 x CO 6MWT) has a higher squared multiple correlation (R2) and a lower standard error of estimate (SEE) and SEE% compared to the equation using 6MWD as the only predictor. These findings will be of interest to clinicians, as well as researchers in the field.

Authors response: We are thankful for the Reviewer’s positive comments.

I have following concerns：

Introduction

1. What is the clinical implication of this study? Is it significant that it has been conducted in young adults but not in stroke? If so, I would think a comparison with trends in young adults would be in order.

Authors response: We are thankful for the reviewer’s insightful comment. The rationale for this study was based on reports that the predictive power of 6MWD for aerobic capacity in people with stroke was questionable. While we have previously shown that inclusion of cardio-dynamic data recorded from ICG could improve the predictive power of aerobic capacity in young adults, we believe the response to exercise stress between people with and without stroke may be different, and the regression model could vary. We have included a more detailed explanation for the rationale of our study in the Introduction, as well as inclusion of a paragraph under the subheading “implications of the study” in the discussion (lines 387-403).

Methods

2. Are patients with arrhythmias excluded? There are descriptions of uncontrolled arrhythmias, but what about controlled Af? Additionally, were there any patients with brainstem lesions? There may be some impact because the respiratory and circulatory centers are located.

Authors response: We did not include patients with brainstem injury, although it was not stated in the exclusion criteria. We have now included this in our list of exclusion criteria (please see line 90-91). The patients with uncontrolled arrhythmias were excluded (this was listed at point 5 in the exclusion criteria). If a stroke patient with arrhythmia was stable, the patient would be included in our study. 

Results

3. A patient flow chart of inclusion/exclusion criteria may be helpful to understand the patient’s characteristics.

Authors response: A flow chart of participant enrollment and reasons for exclusion is now included as Figure 1.

4. My greatest concern is that most of the subjects are male. It is mentioned in the limitation, but we need to be very careful about generalizing. Authors should consider presenting male-only results.

Authors response: Indeed, this is a limitation of our study. For reasons unknown to us, the majority of patients admitted to our center were males during the data collection period. We were not able to explain why this is so but have addressed this under the Limitations of the study. Please see line 405-410.

5. I think the p-value in Table 1 should be a specific number, not p>0.05.

Authors response: The specific p values are now included in Table 1.

6. Kidney disease and pulmonary disease were recorded, but there are no data in Table1.

Authors response: The data for kidney disease and pulmonary disease are now included in Table 1.

7. Table 6 “At the end of end”. Is it “At the end of 6MWT”?

Authors response: Thank you for pointing out this typo error. The error in Table 6 is now rectified.

Discussion

8. It may not be appropriate to apply inferences drawn from our data analysis to female patients, however the pattern of response to exercise stress between male and female patients appeared to be similar. The tendency of the pattern of response may be similar, but the degree to which they are different is quite different.

Authors response: Indeed, we are aware of the limitation of gender bias. However, the pattern of response to exercise stress between male and female patients appeared to be similar. Therefore, we included the results of all the male and female participants in the report.

---

## [Decision Letter · Decision Letter 1]

16 Aug 2022

Stroke volume and cardiac output during 6 minute-walk tests are strong predictors of maximal oxygen uptake in people after stroke

PONE-D-22-12074R1

Dear Dr. Jones,

We’re pleased to inform you that your manuscript has been judged scientifically suitable for publication and will be formally accepted for publication once it meets all outstanding technical requirements.

Kind regards,

Shane Patman, PhD

Academic Editor

PLOS ONE

Additional Editor Comments (optional):

Reviewers' comments:

Reviewer's Responses to Questions

**Comments to the Author**

1. If the authors have adequately addressed your comments raised in a previous round of review and you feel that this manuscript is now acceptable for publication, you may indicate that here to bypass the “Comments to the Author” section, enter your conflict of interest statement in the “Confidential to Editor” section, and submit your "Accept" recommendation.

Reviewer #2: All comments have been addressed

Reviewer #3: All comments have been addressed

2. Is the manuscript technically sound, and do the data support the conclusions?

Reviewer #2: Yes

Reviewer #3: Yes

3. Has the statistical analysis been performed appropriately and rigorously? 

Reviewer #2: Yes

Reviewer #3: I Don't Know

4. Have the authors made all data underlying the findings in their manuscript fully available?

Reviewer #2: Yes

Reviewer #3: Yes

5. Is the manuscript presented in an intelligible fashion and written in standard English?

Reviewer #2: Yes

Reviewer #3: Yes

6. Review Comments to the Author

Reviewer #2: Thank you for including me in this process and for having the confidence in letting me review your interesting work. Comments and suggestions for improvement have been met and made and I recommend the article to be published.

Reviewer #3: Dear Authors

The authors did an excellent job addressing my previous comments and their response seems reasonable. The manuscript and associated figures are now more informative. I have no further comments.

7. PLOS authors have the option to publish the peer review history of their article (what does this mean?). If published, this will include your full peer review and any attached files.

Reviewer #2: **Yes: **Henriette Busk

Reviewer #3: No

---

## [Editor Report · Acceptance letter]

19 Aug 2022

PONE-D-22-12074R1 

Stroke volume and cardiac output during 6 minute-walk tests are strong predictors of maximal oxygen uptake in people after stroke 

Dear Dr. Jones:

I'm pleased to inform you that your manuscript has been deemed suitable for publication in PLOS ONE. Congratulations! Your manuscript is now with our production department. 

Kind regards, 

on behalf of

Assoc Prof Shane Patman 

Academic Editor

PLOS ONE